# The First 3 Years: Movements of Reintroduced Plains Bison (*Bison bison bison*) in Banff National Park

## Adam Zier-Vogel and Karsten Heuer *

Banff National Park, Box 900, Banff, AB T1L 1K2, Canada
* Correspondence: karsten.heuer@pc.gc.ca

**Abstract:** We assessed 3 years of post-release movements of a reintroduced plains bison (*Bison bison bison*) population for evidence of anchoring, settling, exploratory and adaptive behavior within a 1200 km² target reintroduction zone in Banff National Park. We first held them in a soft-release pasture for 18 months, then partially constrained their movements with drift fences and hazing trials to discourage excursions from a 1200 km² target reintroduction zone. Their post-release movements were within 13 km of the soft-release pasture for the first 3 months, but management interventions were needed to keep the animals within 29 km of the release site and inside the reintroduction zone for the remainder of the 3-year study period. Bison exploration was high in the first year but decreased thereafter, as did the size of their annual home range. Step lengths did not decrease but the frequency of "surge movements" (step lengths > 4 km in 2 h) did. Fence visits did not decrease over time but the need to herd/haze the bison from other, unfenced boundary areas did. The reintroduced bison seasonally selected for rugged, high-elevation habitat despite being translocated from a flat landscape. Our results suggest wild bison reintroductions to areas of just a few hundred square kilometres are possible without perimeter fencing, so long as good habitat and management interventions to discourage broad movements are in place. Trends suggest such interventions will need to continue in Banff until the bison range can be expanded and/or bison movements are constrained by other forces, such as regulated hunting outside the park.

**Keywords:** anchoring; Banff National Park; *Bison* sp.; conservation; dispersal; elevation; exploration; fidelity; plains bison; range establishment; range expansion; reintroduction; settling; soft-release; step-length; translocation



## 1. Introduction

Wildlife reintroductions are an increasingly popular tool for restoring endangered species to areas where they have gone locally extinct [1]. The opportunities for such reintroductions are limited by the lack of suitably large tracts of habitat and interjurisdictional policy alignment, especially for large mammals like bison that roam widely [2]. Many reintroductions have therefore failed when animals dispersed long distances after release and had to be recaptured or destroyed due to conflicts with humans [3,4]. Even in remote areas, where such conflicts didn't occur, post-release survival was found to be negatively correlated with dispersal distance [5]. The success of many reintroduction efforts, therefore, depends on animals adopting a home range near to where they were released.

Plains bison (*Bison bison bison*) were overhunted and exterminated from the mountainous area that later became Banff National Park (BNP) in the 1870s and 1880s, close to the same time they disappeared from the adjacent Great Plains [6]. The reintroduction of a wild population in BNP was first considered in 1995 when a local bison viewing facility was closed and it was suggested the captive animals, which are native to the area [7], be released [8]. This triggered a wild bison habitat suitability study [9] and extensive stakeholder consultations that showed strong support for a reintroduction but with significant concerns from surrounding agricultural and hunting interests [10]. Parks Canada responded with a

plan to reintroduce a small number of animals into a target reintroduction zone on a 5-year trial basis, and to keep them from venturing eastward from the park where they are not recognized as wildlife [11,12].

Although plains bison historically moved widely [13], the few unfenced conservation herds that exist today are all constrained by surrounding human development [2]. For example, approximately 5000 bison in Yellowstone are limited to the 9000 km$^2$ park, mostly because of agriculture beyond its borders [14]. A similar situation exists for the 200–400 bison in Canada's 3800 km$^2$ Prince Albert National Park (PANP) [15]. Another small (~320 animals) population is tolerated amongst grazing cattle on 1550 km$^2$ of public lands in the Henry Mountains of Utah, but with tight controls on their numbers and range [16].

Bison herds consisting of hundreds of animals have been shown to range across 693 km$^2$ in PANP [17] and 397 km$^2$ in the Northwest Territories of Canada [18], but also have a tendency to move more widely, even where physical and policy barriers exist [19]. Such behavior typically involves dispersing males [5,20] but could involve an entire herd, as happened with the failed wood bison (*Bison bison athabascae*) reintroduction to Jasper National Park [3].

We employed six strategies to dampen such dispersal tendencies in Banff by:

(1) Ensuring there was adequate habitat [9];
(2) Starting with young animals to limit their attachment to the source area [19];
(3) Ensuring all females were pregnant and calved shortly after translocation to better anchor to the new area [20];
(4) Holding animals for 18 months in a soft-release pasture to increase site fidelity [21–25];
(5) Discouraging dispersals with short (0.2–2 km long), 150 cm high, wildlife friendly bison drift fences consisting of five smooth wires along likely exit routes from the reintroduction zone [26];
(6) Physically hazing bison from boundary areas by foot, horseback or helicopter where drift fences were impractical [27].

Perimeter fencing was not an option due to the scale, remoteness and ruggedness of the reintroduction zone, and the controversial nature of fencing a significant portion of a mostly wild national park.

With the exception of three males that dispersed shortly after being released (and had to be recaptured or destroyed and were lost to the project), the above strategies worked: the main Banff bison population (which grew from 16 to 66 animals) remained within the 1200 km$^2$ target reintroduction zone (estimated carrying capacity of 600–1000 bison [9]) and within 29 km of the soft-release pasture with minimal management interventions (38 fence visits and 6 hazing events) over the 3-year study period.

Details on the design and efficacy of drift fences and hazing trials is covered in previous studies [26,27]. Here, we focus on the 3 years of post-release movements of reintroduced female bison (males will be addressed in a forthcoming paper) and examine the extent to which they adopted the target reintroduction zone as their new home range. We did so in four themes:

(1) **Anchoring**, or how animal fidelity to the soft-release pasture and the targeted reintroduction zone changed over time. A similarly designed reintroduction project for European bison (*Bison bonasus*) saw animals venture <8 km of the pasture for the first 6 months [28], whereas reintroduced elk (*Cervus canadensis*) dispersed 8–19.7 km over 2 years, depending on age and sex class, with an inverse relationship between time spent inside the enclosure and dispersal distance afterwards [29]. Based on these results, we expected the Banff bison to remain within 10 km of the soft-release pasture for the first 6 months, and for fencing and hazing interventions to limit their range to <30 km from the soft-release pasture thereafter. We expected the need for such interventions to decrease with time as the animals learned the boundaries of the target reintroduction zone, and an initially high return rate to the soft-release pasture to wane after the first year with lower rates of return afterwards.

(2) **Settling**, or how bison behavioral states, as measured by step lengths and turning angles, shifted over time [30,31]. We expected bison to spend a higher proportion of their time in a "travelling" state immediately after release (i.e., longer step lengths with straighter paths) and shift to "feeding-resting" states (i.e., shorter steps with more frequent turns) as they adjusted to their new surroundings. Step lengths for an established wild plains bison population in Prince Albert National Park averaged 70 m per hour [32], so we expected bi-hourly step lengths to be initially elevated immediately after the release of the Banff bison (>200 m every 2 h) and to then decrease as the animals grew accustomed to their new home range. We also expected large surges (i.e., step lengths > 4 km in 2 h) to be rare, and to decrease with time.

(3) **Exploration**, or the rate at which bison ventured into new, previously unvisited areas. A study of reintroduced European bison [28] recorded multiple, discrete pulses of exploration, with very high rates in the initial days, followed by reduced rates as newly familiar areas became bases from which the animals staged further exploration. Exploratory pulses continued to the end of their 6-month study period. Other studies of reintroduced ungulates showed exploration and home-range establishment occurring in distinct phases and within a wide range of timescales. For example, elk reintroduced to the Missouri Ozarks transitioned from a dispersive phase to a home-ranging phase after only 10 days [21] while elk introduced to the Bancroft region of Ontario took 1–3 years before settling into home-ranging movements [29]. Based on these trends, we expected exploration to be high for the Banff bison in the first week, and then to pulse upwards several times for up to a year before stabilizing at low levels. We expected the home-range size to similarly shrink and stabilize within a year.

(4) **Adaptation**, or the tendency for bison to explore and exploit the novel, rugged, high-elevation mountain habitat versus the valley bottom meadows that are like the flat, low-elevation habitat they were translocated from in Elk Island National Park. The theory of natal habitat preference induction (NHPI) predicts dispersing animals will select similar areas to those they came from to minimize risk of assessing unfamiliar habitats [4]. We, therefore, expected the Banff bison to initially prefer low-elevation, valley-bottom meadow habitats and not to explore higher elevation habitats until a year or more had passed.

## 2. Materials and Methods

### 2.1. Study Animals and the Soft-Release Pasture

Six males and ten female plains bison, all aged between 2 and 3 years old, were translocated from Elk Island National Park to an 18 ha. soft-release pasture in the BNP backcountry in February 2017, where they were held for 18 months (Figure 1b). Although deemed good bison habitat for all but high snow winters [9], the soft-release pasture site was primarily chosen for its centrality within the 1200 km$^2$ reintroduction zone (Figure 1) and its existing building infrastructure. Founding females were pregnant and gave birth to ten calves in spring 2017 and an additional five calves in spring 2018 before all animals (N = 31) were released at the end of July 2018. Additional calves were born later that summer and each summer afterwards, and the herd grew to 66 animals by the end of the study period. Vegetation in the pasture is representative of the greater reintroduction zone, but, due to the small size of the pasture, animals were fed hay while in captivity. No feed was provided after the animals were released. Other than infrequent splitting into small groups (typically during calving events), all bison cows and their young remained together throughout the study period, forming what we refer to as the "main herd."

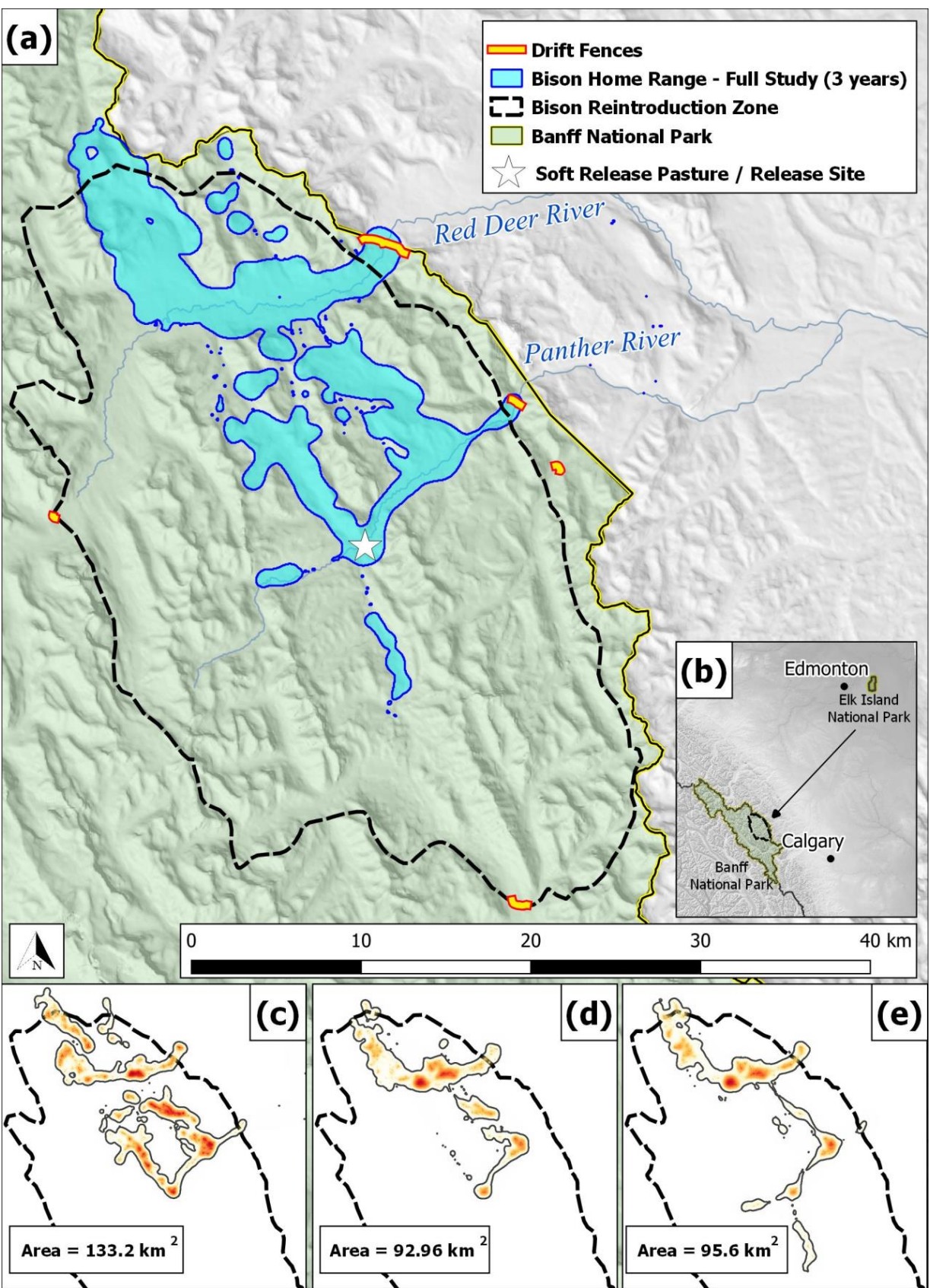

**Figure 1.** Home range of a reintroduced bison herd (cows and their young) in the first three years of free roaming. Full study (**a**), study years 1–3 (**c**–**e**) with Brownian bridge utilization distribution and 99% isopleths. (**b**) Translocation route.

*2.2. Data Collection*

Between 4 and 9 bison cows carried GPS radio collars (Vectronic Aerospace) throughout the 3–year (1096-day) study period (29 July 2018 to 28 July 2021). Collars were first fitted to bison at the end of the soft release holding period and were subsequently replaced if they dropped off before the end of the study period, either by helicopter net gun capture (*n* = 3, October 2019 and January 2020) or by free-range dart from horseback (*n* = 4, July and September 2020). Collar fix rates were mostly at 2 h intervals with a minority at 7 h intervals for the first 2 months of the study period. We filtered GPS data to include only 3D fixes with a minimum of 4 satellites, and removed all locations outside of normal parameters (speeds > 80 km/h and anomalous locations), and time periods in which the collars were active but not attached to animals. Images from 12 remote cameras distributed in a 10 km$^2$ grid and frequent field observations corroborated collar data, but were not used for analysis. A total of 83,420 GPS locations were analyzed.

*2.3. Analyses*

2.3.1. Anchoring

Mean daily Euclidean distances to the soft-release pasture were calculated from pooled collar data, directions to the soft-release pasture determined for each GPS location, and the herd's returns to the soft-release pasture were tallied over the study period. Returns occurred when half or more of the collared animals approached within 2 km of the soft-release pasture. This approach excluded events involving single individuals or small subgroups unrepresentative of the main herd.

We tallied how often management interventions (drift fence encounters, hazing events and helicopter captures) were required to keep the bison within the intended reintroduction zone. Hazing events occurred when staff intentionally moved bison by foot, horse, or by helicopter.

2.3.2. Settling

We looked for evidence of a settling trend in bison movements by investigating step-length data (distance travelled between consecutive fixes) in the following ways: bi-hourly and daily hidden Markov model (HMM) movement states (combination of step lengths and turning angles), bi-hourly step lengths, daily maximum step lengths, daily displacements, total daily travel distances, and frequency of "surge movements". Only data acquired at 2 h fix intervals (N = 81,045) were used for these calculations to eliminate bias from different fix rates, with the exception of daily displacements and daily HMM states, which were derived from the full dataset. Turning angle, step lengths and HMM states were calculated using the moveHMM package (version 1.7) [33] in R Studio. Medians for bi-hourly step lengths and daily maximum step lengths were determined from pooled collar data. Daily displacements (change in average position over a 24 h period) was determined by calculating the mean position for each individual for each day, the distance between those positions, and then the mean for all individuals. Total daily travel distances (sum of all movements within a day) were calculated as the sum of all consecutive step lengths within a 24 h period for each individual, and then the mean for all individuals by day. Surge movements were defined as bi-hourly step lengths larger than 4 km, which were anomalously large amongst normal daily movements. HMM state and surge movement results were calculated proportionately rather than as raw counts to account for the varying number of collars deployed through the study period.

2.3.3. Exploration

We overlaid a grid of 500 m wide hexagons on the reintroduction zone and defined entry into a cell by any of the collared bison for the first time as an exploration event. This grid size was derived on a trial and error basis, taking into account the average meadow habitat patch sizes in the reintroduction zone while balancing detection sensitivity and

stochasticity. Home ranges were calculated with the BBM package in R Studio [34] and defined as the area contained within Brownian Bridge 99% probability isopleths [35,36].

### 2.3.4. Adaptation

Elevations were determined for all collar fixes and plotted by extracting values from a 30 m resolution digital elevation model. Mean daily and monthly elevations were calculated from pooled collar data.

### 2.3.5. Statistical Tests across Themes

We used the Kruskal–Wallis rank-sum test to identify differences between study years [37] and, where applicable, Dunn's test with Bonferroni correction for post hoc analyses [38]. For proportional results, we used the Wilson score method to compare between years [39] and used pairwise comparisons of proportions with Bonferroni correction for post hoc tests. A significance of 0.05 was used for all tests.

## 3. Results

### 3.1. Anchoring

The bison moved farther than anticipated upon release (6.5 km in the first few hours), but their movements slowed to remain within 12 km of the soft-release pasture for the next 7 months (Figure 2). They encountered their first drift fence halfway through this period. Distance from the soft-release pasture increased thereafter as the bison explored a new (Red Deer) valley during the eighth month. Distance from the soft-release pasture reached a maximum of 29.6 km at the 11-month mark when the herd had to be hazed from the northern edge of the reintroduction zone. Distance from the soft-release pasture remained below this maximum for the remaining 25 months of the study period.

The herd returned to the soft-release pasture on 13 occasions (Figure 2). Half of these returns (N = 6) occurred within 5 months of the release, after which no returns occurred until the second (N = 3) and third years (N = 4). Annually, 5.2% of all collar fixes were within 2 km of the soft-release pasture in the first year, dropping to 3.5 and 2.7% in the second and third years. All returns occurred in the fall and early winter (between September and January), and averaged 73.1 h long and ranged from 12 to 234 h.

Animal movements were constrained by drift fences on 33 occasions, (13, 5, and 15 times in the first, second, and third years, respectively). Herding events occurred on 12 occasions (7,4, and 1 by year, respectively), some of which coincided with fence visits (Figure 2). Altogether, the bison were prevented from leaving the reintroduction zone on 39 occasions (either by fences, herding, or a combination of both), a total of 16, 7 and 16 events by year, respectively.

Bison movements were highly asymmetrical: 44, 38, and 14% of fixes occurred to the NW, N, and NE of the release site, respectively. Movements in all remaining directions account for only 3% of all fixes (Figure 1a).

### 3.2. Settling

Two movement states were identified with HMMs for 2 h steps: a "feeding-resting" state with low mean step lengths (141 m) and high tortuosity (0.1 turning angle concentration) and a "travelling" state with high mean step lengths (817 m) and relatively straight paths (1.1 turning angle concentration). Overall, the bison spent most of their time (84.2%) in the settled state and did so consistently between study years ($\chi^2$ = 3.23, df = 2, $p$ = 0.20).

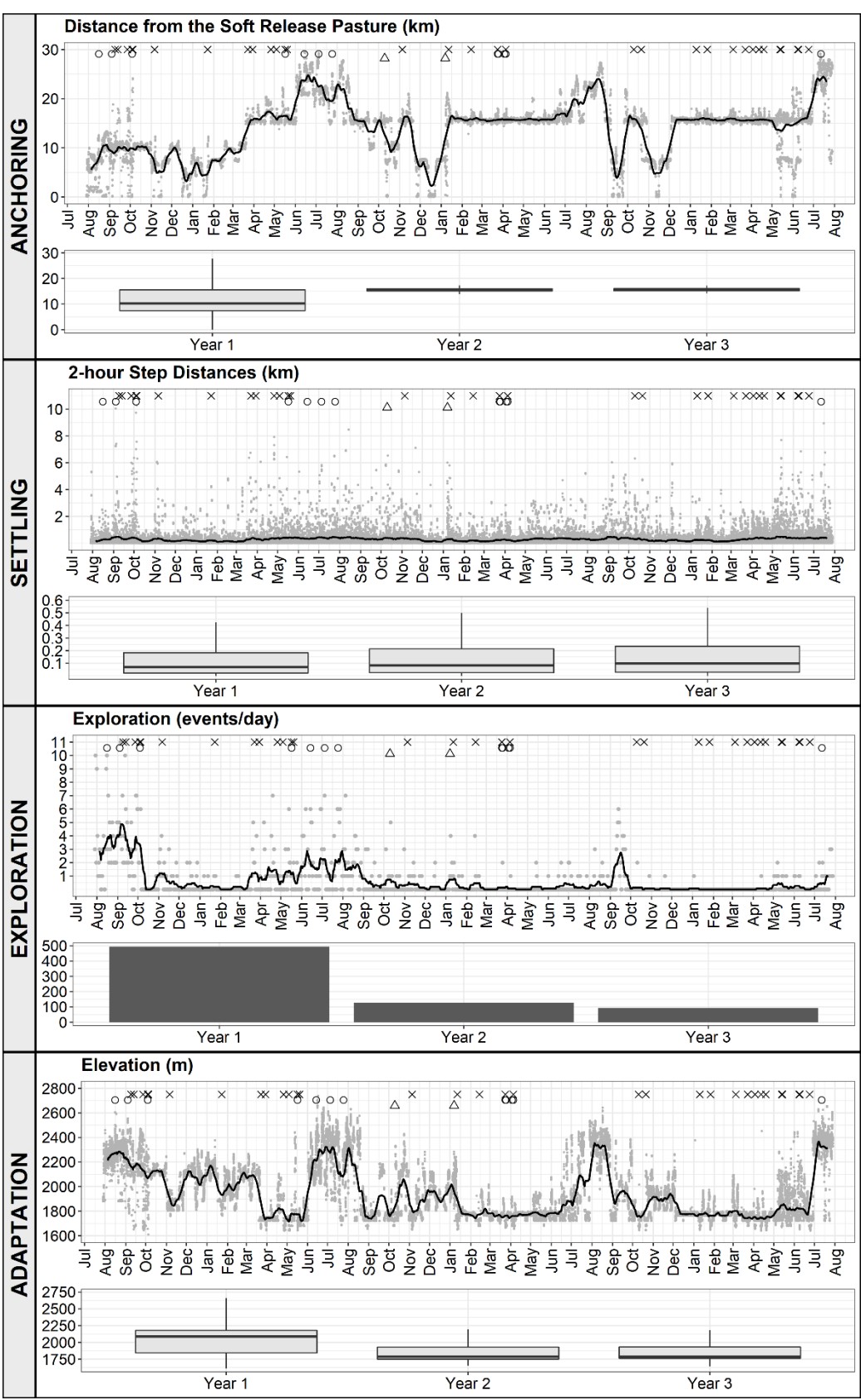

**Figure 2.** Reintroduced bison movements in 4 themes. Anchoring, settling, and elevation boxplots depict median, 1st quartile, 3rd quartile, min and max (outlying values omitted). Exploration bar chart summarizes exploration events/year. Chart legend: ● = individual fixes. ⌒ = 30-day moving average. × = drift fence encounter. ○ = herding event. △ = helicopter capture.

Application of HMMs at a coarser daily scale identified two different movement states: a "feeding-resting" state with mean displacements of 1.1 km/day and high tortuosity, and a "travelling" state with mean displacements of 2.1 km/day and relatively straight paths. The bison split their days roughly equally between the two states, slightly favoring the feeding-resting state at 55.1% overall. The proportion of time spent in each state was consistent across the study years ($\chi^2 = 2.40$, df = 2, $p = 0.30$).

Bi-hourly step lengths were typically short (median = 107 m) and were significantly different between the three study years ($\chi^2 = 295.27$, df = 2, $p < 0.001$) with the median increasing annually (89, 110, and 126 m, $p < 0.001$ for all post hoc tests).

Daily maximum bi-hourly step lengths had an overall median of 918 m with medians increasing (861, 925, and 1020 m, respectively) and differing between study years ($\chi^2 = 6.39$, df = 2, $p = 0.04$), although only the increase between year 1 and 2 was significant ($z = -0.40$, $p = 1.0$).

Total daily travel distances had a median of 2.7 km, and were significantly different between study years ($\chi^2 = 13.00$, df = 2, $p = 0.002$) with annual medians of 2.3, 2.8, and 2.9 km. Only the increase between the first and second study year was statistically significant ($z = -2.1$, $p = 0.02$)

Surge movements accounted for only 0.3% of all 2 h step lengths and occurred on only 6.4% of days. The proportion of surge movements was significantly higher in the first study year (0.47%), when compared to the second and third study years (0.23 and 0.25%, respectively; $\chi^2 = 30.485$, df = 2, $p < 0.001$). Of the surge movements, 33.2% occurred within 48 h of management interventions but the decreasing pattern remained when these intervention periods were removed from the analysis (0.31, 0.24, and 0.16% by study year).

### 3.3. Exploration

The bison's initial rate of exploration was high with 35.6% of their total 3-year range explored in the first 69 days after release (Figure 2). A second, less intense peak of exploration occurred at 8 months, when they ventured into a new, major (Red Deer) valley for the first time and spent 244 days (March through October) exploring a further 38.3% of their 3-year range. Most (69.1%) exploration events occurred in year 1, dropping to 18.0% in year 2 and 12.8% in year 3. Overall, 48.6% of all explorations occurred in the months of August and September; however, this is almost certainly influenced by the timing of the release in late July. Looking at data from the second and third years alone, explorations remained concentrated (68.2%) in the months of July (12.8%), August (29.0%), and September (26.4%). Exploration rates in the second and third years were lowest in the early spring (0.8% in March and 0.9% in April).

Annual home-range sizes for the first 3 years were 133.2 km$^2$, 92.9 km$^2$, and 95.6 km$^2$, respectively. Only 50.3% of the area visited in the first year was visited again in the second year, while 74.0% of the area visited in the second year was re-visited in the third year.

### 3.4. Adaptation

The bison climbed from the bottom of the valley at the soft-release pasture (1880 m) to nearby high elevation slopes (~2300 m) within hours of their release, then stayed above 2088 m (ASL) throughout their first 3 months of freedom (Figures 2 and 3). Monthly mean elevations dropped to 1750 m the following fall and winter before increasing again in the spring (2210 m). The bison followed a similar seasonal elevation pattern each year thereafter, preferring low elevations in winter and higher elevations in summer (Figure 2). Monthly mean elevation ranged from 1765 to 1952 m in the snow-bound months (October–May) and from 1977 to 2208 m in the snow-free months (June–September).

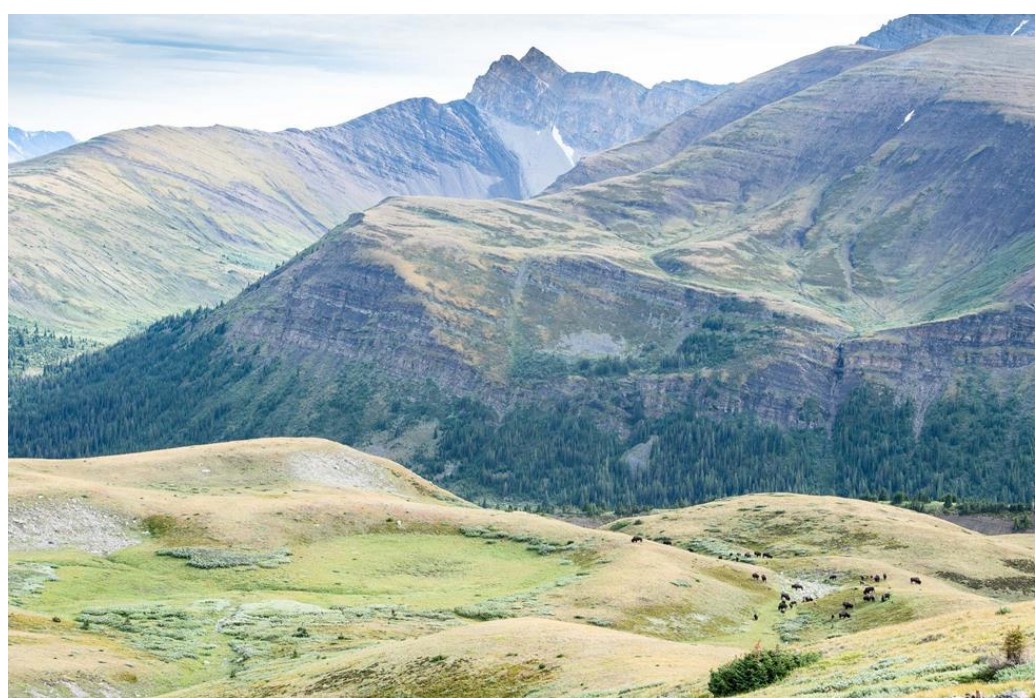

**Figure 3.** Regardless of the availability of flat, valley-bottom meadows, the bison gravitated immediately to rugged, high elevation habitats, despite being translocated from a flat, low-lying environment (K. Heuer/Parks Canada).

## 4. Discussion

Our bison reintroduction project, which was constrained to a 1200 km$^2$ target reintroduction zone, required several strategies to discourage bison movements beyond its boundaries. These included: ensuring adequate habitat existed within the area; starting with mostly young, pregnant animals that calved shortly after translocation; holding animals for 18 months in a soft-release pasture; and discouraging dispersals with bison drift fences and active hazing when bison entered boundary areas. We assessed the efficacy of those strategies in four themes.

### 4.1. Anchoring

The soft-release pasture did not play as much of a role in anchoring bison to the area as we expected, with no returns in the first 3 months after the bison were released, and infrequent returns thereafter (Figure 2). Unlike other studies [21], our soft-release pasture did not function as a base from which the animals staged further explorations but was one of many spatially disparate areas of concentrated use that shifted north to the next valley by the end of the first year (Figure 1c–e). Directional movements were highly asymmetrical from the release site as a result, occurring almost exclusively in a northerly direction (Figure 1). Although the soft-release pasture remained part of the bison's home range throughout the 3-year study period, they spent less time in the area with each passing year (Figure 2). The returns that did occur all happened between September and January, when forage quality is reduced due to senescence and snow cover becomes common. The seasonal return of bison at this time may have been associated with memories of being fed hay at that location while in captivity.

Although impossible to verify experimentally, the low bison dispersal distances after the release could be attributed to a the 18-month holding period. The failed bison reintroduction in Jasper, which also translocated animals from Elk Island into the mountains, held bison in a soft-release pasture for only 43 days. Those animals dispersed long distances in a short time: 10 kms within 2 days of being released, 32 kms within 5 days, and nearly 200 kms when they were recaptured a month later [3].

Drift fences and hazing events, which we collectively refer to as management interventions, played an important role in limiting bison dispersal beyond our 1200 km$^2$ target reintroduction zone, especially in the eastern part of the Red Deer Valley, where 69.2% of these interventions occurred (Figure 1). Overall, the main herd's movements were restricted 35 times to the east and 4 times to the north by our interventions, and, contrary to our expectations, did not wane with time. Instead, they more than doubled in frequency between year 2 and year 3, indicating the bison had not learned or accepted this boundary, and/or were persistently motivated to explore beyond it for reasons below.

Of the events at the Red Deer boundary, 60% occurred between March and May, a time when snow can still be deep, and animals are in the poorest body condition from the combined effects of a lean winter and the building demands of spring parturition. Lands immediately to the east of the reintroduction zone, which comprise some of the best ungulate winter range in Alberta [40] are lower, warmer, not as snowy, and tend to green up several weeks before those in the reintroduction zone, which likely explains why the bison persist in moving that way at that time of the year.

*4.2. Settling*

Bison step lengths did not decrease over time as we expected but increased with each successive year after the release. This may be due to step length not being a good indicator of how settled an animal feels or is indicative of a prolonged adjustment period. Indeed, reintroduced elk in one study took up to 3 years to shift from a "dispersive" state into a more stable "home ranging" state [29]. Regardless of the trend, it is important to note that movement rates were lower for Banff bison than those reported for other bison populations. For example, mean step distances were 140 m/h in Banff compared to a 243 m/h mean found in an established population of wild Plains bison in Prince Albert National Park [32]. Mean 24 h bison displacement in Banff (1.5 km/day) was also lower than for bison in Prince Albert National Park (5.8 km/day) [32], and for Yellowstone National Park (3.2 km/day) [41], but larger than for reintroduced European bison in Germany (medians ranging from 389 to 900/day) [28].

As predicted, the frequency of surge movements (>4 km in 2 h) decreased over time, even when those associated with management interventions (33.2%) were removed from the analysis. This downward trend may be due to fewer hazing or helicopter captures as the study progressed. It may also indicate the animals habituated to sights, sounds and other stimuli that were initially novel once they grew accustomed to the reintroduction zone.

*4.3. Exploration*

The rate at which the bison ventured into new, previously unvisited areas did not change linearly, but, as predicted, unfolded in pulses. The greatest spike occurred right after the release, when everything was new, before waning for several months as the bison "camped" in a small tributary valley. The following spike occurred at the 8-month mark, when the animals pushed into a new (Red Deer) valley that became the central node for subsequent movements and much smaller bouts of further exploration (Figure 2). All these spikes coincided with movements into new valleys, both large and small. This could be attributed to new viewsheds opening up before the bison and invoking further exploration, or new watercourses with new wildlife trails that naturally pulled bison onwards until they discovered the next grassy meadow.

Although much diminished, exploration continued at low rates after the first year, primarily in summer and fall, presumably because good forage was widely available in those seasons, and snow cover did not impede movement or feeding. This all happened against a background of management interventions that allowed for exploration within the limits of the target reintroduction zone. Had bison movements not been constrained by fences or hazing events, the pattern of exploration and range establishment would likely have been very different, maybe so different that, like the Jasper project which lacked such interventions, the project would have failed [3].

*4.4. Adaptation*

Bison are known to be exceedingly adaptable [42], but we were nonetheless unprepared for their immediate adoption of some of the most rugged, high-elevation habitat the reintroduction zone had to offer, especially given that they originated in the flat, low-lying parkland of Elk Island. This contradicts NHPI theory, which posits that animals in new areas will gravitate to similar habitats from where they came in order to reduce mistakes [4]. Whether it was due to improved forage quality at higher elevations [43,44] or to escape summer heat and insects [45], is unknown, but such seasonal altitudinal migrations, which have also been recorded for bison in the Henry Mountains of Utah [46], were repeated every summer over the 3-year study period for Banff bison (Figure 2). Such adaptability facilitates the adoption of a target zone as a reintroduced animal's new home range.

**5. Conclusions**

The first 3 years of BNP's bison reintroduction has been a success: initially high rates of exploration and occasional large movements significantly decreased as the size of the animals' annual home range contracted and stabilized, and management interventions, such as wildlife-friendly drift fences, and hazing animals from peripheral areas, have helped keep them in the 1200 km$^2$ target reintroduction zone and out of an adjoining jurisdiction where they are not considered to be wildlife. However, the need for such interventions increased, rather than decreased, over the 3-year study period, which suggests they will need to continue, especially as the population grows, until bison range can be expanded and/or bison movements become constrained by other forces, such as regulated hunting outside the park. This study demonstrates that wild bison reintroductions without perimeter fencing are possible in areas of just a few hundred square kilometres, so long as good habitat and management interventions to discourage broader movements are in place.

**Author Contributions:** Conceptualization, writing, supervision and project administration, K.H.; methodology, formal analysis, writing, data curation and validation, A.Z.-V. All authors have read and agreed to the published version of the manuscript.

**Funding:** This research was funded by Parks Canada.

**Institutional Review Board Statement:** The animal study and capture protocol was approved by Parks Canada's Animal Care Task Force under a 2016 Parks Canada Research and Collection Permit.

**Data Availability Statement:** Not applicable.

**Conflicts of Interest:** The authors declare no conflict of interest.

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
