# Peer review of "The First 3 Years: Movements of Reintroduced Plains Bison (Bison bison bison) in Banff National Park"

_diversity, doi:10.3390/d14100883_

Round 1

Reviewer 1 Report

I read with great interest the manuscript entitled “The First 3 Years: Movements of Reintroduced Plains Bison (Bison bison) in Banff National Park”. In my opinion, the work is very interesting and important in the context of real problems related to the reintroduction of large mammals. However, I notice many shortcomings, especially in the methods and results. Detailed comments below

lines 73-74- point 6- grammatically inconsistent with the introductory sentence (We employed 5 strategies to dampen such dispersal tendencies in Banff, which consisted of ... Where drift fences were impractical, physically hazing bison from...), I suggest rewriting

line 85- rather "in previous studies" than "elsewhere"

lines 91-131 - In my opinion this description should be part of the methods, It should only be mentioned in the introduction. Alternatively, in the introduction, you can describe the methods of assessing the first years of reintroduction with the cited literature and mention it later in the methodology. The aim of the study should also be clearly indicated here.

168-169 - why half or more of the collared animals? Why not each animal (collar) was treated separately?

lines 201-215 - the description of statistical analyses in not clear. Please indicate how many models were built, what was the dependent variable in each model (how expressed - e.g. km), how the model assumptions were verified (at least for linear models), what were the explanatory variables (both continuous and categorical - for them indicate groups), and how they were expressed (as above), for generalized model indicate the distribution and link function used.

Also, for each model number of observations used should be indicated. I understand that full model was compared to the null model.

Why year was not included in the model? It should be as a categorical variable, then you could also add interaction to verify how it differed in each year.

Results: I my opinion too long descriptive part. Please shorten the description and focus on more important parts. Please add tables for each model (as supplementary material) indicating the parameter estimates. What is the delta AIC here? with null model? What is here Akaike weight? How it was calculated? I mean, what represent this measure? I guess that is just for explanatory variable (if in the model there is one explanatory variable Akaike weight will always be  1).

232-233- I don't understand how it was calculated?

Discussion: It should start with the introduction part, where concise information on the aims of the study and main findings.

Conclusions: too many descriptions. Conclusions should be based on the results and not repeat them.

Author Response

Dear Reviewer 1,

Your review and comments are appreciated and will make the paper better. Thank you. 

Most of your points were consistent with those of our other reviewer but a couple were contradictory. We did our best to address them, as described (in bold) below:

lines 73-74 (now 78-79)- point 6- grammatically inconsistent - Rewritten;

line 85 (now 92)- rather "in previous studies" than "elsewhere" - Replaced;

lines 91-131 (now 93-151) - In my opinion this description should be part of the methods. - This is contrary to Reviewer 2's comments, who liked it. We retained it for the following reasons: 1) It lays out the study structure early in the paper; 2) It provides background on each theme from the available literature; 3) It lays out our expectations clearly for each theme. For these reasons, we do not feel  it is appropriate for Methods. 

168-169 -(now 198-202) why half or more of the collared animals?  We added the following justification: "This approach excluded events involving single individuals or small subgroups unrepresentative of the main herd".

lines 201-215 (now 249-262)- the description of statistical analyses is not clear. We reassessed our analyses and decided to exclude the linear models and AICs to shorten and clarify the paper. To be truthful, they didn't add anything (or change) the narrative told by the descriptive statistics alone.  We also included information about management interventions into the timelines in Figure 2 to be more transparent. 

Results: Please shorten the description and focus on more important parts.  Please add tables for each model (as supplementary material). We shortened this section significantly. Excluding the models helped remove noise. Supplementary tables with model results are no longer relevant or necessary. 

232-233- I don't understand how it was calculated? All models and AIC references have been removed.

Discussion: It should start with the introduction part, where concise information on the aims of the study and main findings. We added a few sentences to the start of the Discussion to address this shortcoming. 

Conclusions: too many descriptions. Conclusions should be based on the results and not repeat them. We shortened and clarified Conclusions accordingly.

Thank you again for your careful reading of our submission, and your constructive comments. 

Karsten Heuer

Reviewer 2 Report

Nice study - I like the formal preditions that were set up realting to the different aspects that enabled you to undertake a concise and focussed assessment of post-release performance.

I've annoted the ms pdf and so wont repeat it all here.

In brief I had some queries about the original site selection, site carrying capacity, natural patterns of bison migration/seasonl movements, and a request for some more edplnatory detail in places. Your presentation of AIC model selection seems in error - I think you've given raw AICc values, rather than delta AIC (which would be 0 I suspect, given the weights).

Nicely presented and well-written study.

Pleasure to read.

Author Response

Dear Reviewer 2,

Your review and comments are appreciated and will make the paper better. Thank you.

I've replied to specific comments in the attached annotated MS pdf, as you provided them. Hopefully this is the most convenient format for you. Cross referencing with the revised MS will be necessary.

Sorry for the errors in our presentation of AIC model selection. We reassessed our analyses and decided to exclude the linear models and AICs to shorten and clarify the paper. To be truthful, they didn't add anything (or change) the results or narrative told by the descriptive statistics alone, even once we had corrected everything.  We also included information about management interventions into the timelines in Figure 2 to be more transparent. See revised MS.

Thank you again for your careful reading of our submission, and your constructive comments.

Karsten Heuer

Round 2

Reviewer 1 Report

In my opinion, the manuscript has been significantly improved. It was a good decision to remove models that ultimately contributed little to the entire work and made it information difficult to perceive. The more so because the methodology was not clear.

I have no detailed comments.